# CRISPR/Cas-Based Techniques for Live-Cell Imaging and Bioanalysis

**DOI:** 10.3390/ijms241713447

**Published:** 2023-08-30

**Authors:** Shuo Huang, Rui Dai, Zhiqi Zhang, Han Zhang, Meng Zhang, Zhangjun Li, Kangrui Zhao, Wenjun Xiong, Siyu Cheng, Buhua Wang, Yi Wan

**Affiliations:** 1College of Life Sciences, Hainan University, Haikou 570228, China; 20203104896@hainanu.edu.cn (S.H.); 20213001574@hainanu.edu.cn (Z.Z.); zhh18639149198@163.com (H.Z.); 20213001597@hainanu.edu.cn (M.Z.); 20213001528@hainanu.edu.cn (Z.L.); 15375487439@163.com (K.Z.); 13324554537@163.com (W.X.); 2Institute of Oceanography, Hainan University, Haikou 570228, China; dr202209@163.com; 3College of Art and Design, Hainan University, Haikou 570228, China; 20213004765@hainanu.edu.cn; 4State Key Laboratory of Marine Resource Utilization in South China Sea, Hainan University, Haikou 570228, China

**Keywords:** CRISPR/Cas, bioimaging, bioanalysis, nucleic acid analysis, protein analysis

## Abstract

CRISPR/Cas systems have found widespread applications in gene editing due to their high accuracy, high programmability, ease of use, and affordability. Benefiting from the cleavage properties (*trans-* or *cis-*) of Cas enzymes, the scope of CRISPR/Cas systems has expanded beyond gene editing and they have been utilized in various fields, particularly in live-cell imaging and bioanalysis. In this review, we summarize some fundamental working mechanisms and concepts of the CRISPR/Cas systems, describe the recent advances and design principles of CRISPR/Cas mediated techniques employed in live-cell imaging and bioanalysis, highlight the main applications in the imaging and biosensing of a wide range of molecular targets, and discuss the challenges and prospects of CRISPR/Cas systems in live-cell imaging and biosensing. By illustrating the imaging and bio-sensing processes, we hope this review will guide the best use of the CRISPR/Cas in imaging and quantifying biological and clinical elements and inspire new ideas for better tool design in live-cell imaging and bioanalysis.

## 1. Introduction

CRISPR is a sequence discovered within prokaryotes, and when combined with Cas proteins, it forms a defense system that counters the proliferation of viruses throughout the evolutionary history of life [1]. CRISPR/Cas systems employ CRISPR RNA (crRNA) to accurately direct the recognition and targeting of foreign nucleic acids. This mechanism equips bacteria and archaea with a form of adaptive immunity, enabling them to effectively defend against viruses [2]. Recently, the discovery of CRISPR/Cas systems has promoted the booming development of molecular biology and gene-editing domains [3]. Scientists harness the power of CRISPR/Cas systems to develop genome editing and multiple technologies across various organisms, including humans [4], animals [5,6], and plants [7]. CRISPR/Cas systems are harnessed for their precision, user-friendly operation, and cost-effectiveness [8]. As a result, researchers are actively exploring new avenues to exploit the application scope of CRISPR/Cas systems [9]. For example, by utilizing dCas proteins, which lack endonuclease activity and exhibit target specificity, live-cell imaging becomes feasible with CRISPR/Cas systems [10,11]. Moreover, the CRISPR/Cas systems also have broad application prospects in bioanalysis [12]. In this review, we emphasize the research advances of CRISPR/Cas systems in bioimaging and bioanalysis and provide ideas for the further improvement of CRISPR/Cas (Cas9, Cas12a, Cas13, and Cas12f)-based imaging and analysis.

### 1.1. The Mechanism of the CRISPR/Cas9 System

The CRISPR/Cas9 system is a natural immune system found in prokaryotes that fights off the invasion of viruses [13]. The system is composed of three key elements: CRISPR, which consists of repetitive DNA sequences; Cas9, an endonuclease responsible for DNA cleavage; and the Cas1-Cas2 or Cas4 proteins, vital components for gathering and storing viral DNA (as illustrated in Figure 1A) [14]. When a bacterial cell is invaded by a virus, it integrates a portion of the viral DNA into its own CRISPR region, forming new spacer sequences. This process is mediated by Cas1-Cas2 or Cas4 proteins. Through this mechanism, bacteria are able to acquire and store genetic information about the virus, which is then incorporated into CRISPR RNA (crRNA) [15]. The crRNA forms a complex with the Cas9 enzyme. These complexes can bind to the corresponding viral DNA sequences, allowing Cas9 to cleave the DNA and prevent viral replication and infection [16]. Harnessing this innate defense mechanism, researchers have unlocked the potential of the CRISPR/Cas9 system for gene editing. By customizing the crRNA to match the specific DNA sequence of a target gene and guiding the CRISPR/Cas9 complex to that gene, precise cutting and editing of the desired gene can be accomplished [17]. In addition to applications in gene editing, CRISPR/Cas9 can be used to label specific cells or molecules. By combining the Cas9 protein with fluorescent tags or other imaging probes, the Cas9 protein can interact with the target sequence, and the labeled targets can be observed through imaging techniques such as fluorescence microscopy. This approach is referred to as CRISPR imaging [18]. Furthermore, CRISPR/Cas9 is also used in the field of bioanalysis in nucleic acid analysis, protein analysis, small molecule analysis, cell analysis, and other aspects [19,20,21,22]. This review provides an important basis for the future development of novel bioanalytical and bioimaging tools based on CRISPR/Cas technology.

### 1.2. The Mechanism of the CRISPR/Cas12 System

Cas12a, alternatively known as Cas12a, is an enzyme guided by RNA that plays a significant role in the adaptive immune system of bacteria. As a key component of the CRISPR/Cas12a system, it recognizes specific DNA sequences and performs cleavage, contributing to the organism’s immunity against invasive genetic elements. Cas12a processes its own crRNA, allowing it to cleave both single-stranded and double-stranded DNA. Additionally, Cas12a requires a specific target recognition site known as a PAM sequence, which can be either TTTN or UUUN (Figure 1B) [23]. However, Cas12a utilizes a single RuvC domain guided by crRNA to catalyze its cis- or trans-cleavage effect on DNA. This unique interference mechanism enhances its applications in genome editing [24]. Through the fusion of Cas12a protein with fluorescent probes, the nucleolytic activity of Cas12a protein can be utilized to trigger the release or signal amplification of the probes, resulting in the generation of a fluorescent signal. Furthermore, by further harnessing the gene-editing capability of Cas12a, one can indirectly achieve biological imaging [25].

In 2018, the smallest CRISPR/Cas system, Cas12f (Cas14), was discovered [26]. The Cas12f enzyme exhibits precise DNA cleavage capabilities, making it valuable for genome and gene editing. Unlike Cas9 and Cas12a, which need the help of accessory proteins for guidance, Cas12f is guided by a single RNA molecule. This simplified mechanism enables Cas12f to directly target genomic DNA, simplifying its design and application [27] (Figure 1C).

### 1.3. The Mechanism of CRISPR/Cas13 System

The CRISPR/Cas13 system recognizes and cleaves the target RNA without the requirement of a PAM site, facilitated by the complementary binding between the crRNA and the target RNA (Figure 1D) [28]. Detection methods based on CRISPR/Cas13 have been developed to differentiate virus strains and perform fine-typing analysis, showcasing the versatility of this system [29]. Cas13a, also known as C2c2, is the first identified isoform of Cas13. Unlike Cas9 and Cas12, Cas13 specifically targets RNA molecules through RNA-directed RNA endonuclease activity [12]. When Cas13 binds to the target RNA, its enzymatic activity is activated, leading to the release or amplification of a fluorescent probe, thereby generating a fluorescent signal. By detecting these fluorescent signals, imaging of the target RNA can be achieved [30].

## 2. Application of CRISPR in Live-Cell Imaging

The imaging of chromatin [31], genomic loci [11], RNA [32], and proteins in live cells is crucial for studying their relationships and coordinated regulation [32,33]. CRISPR/Cas-based imaging systems utilizing deactivated Cas9 (dCas9) and deactivated Cas13 (dCas13) have been established [10]. In this section, we present a detailed analysis of the CRISPR/Cas-based imaging systems, which enable real-time visualization of proteins, RNA, and genome/chromatin (Table 1 and Figure 2). We thoroughly evaluate the strengths and limitations of CRISPR/Cas-based imaging technology. Furthermore, we review the most recent developments in CRISPR/Cas-based techniques and discuss the difficulties and fixes.
ijms-24-13447-t001_Table 1Table 1Advantages and limitations of the CRISPR/Cas-based imaging methods.ApplicationImaging SystemCas ProteinEffector Molecule(s)Advantage(s)Limitation(s)Ref.Chromatin/genomic loci imagingFP-baseddCas 9dCas9-EGPFSimple and versatile tool for imagingLow SNR, poor labeling efficiency[31]dCas 9dSaCas9-EGFPdSpCas9-mCherryAchieved two-color CRISPR imagingRelatively weak fluorescence signal and low signal-to-noise ratio[34]dCas 9sgRNA-aptamer-FPSimultaneous labeling of six genetic lociComplex sgRNA engineering, efficient delivery system needed[35]dCas 9dCas9-Suntag-scFv-GCN 4-sfGFPAmplification of fluorescence signal intensity to improve signal-to-noise ratio enables labeling of low-repeat motifsLarge array size affects Cas9 functionality[36]dCas 9dCas9-SuTag10×-sfGFP-LEXYSimultaneous labeling of nine duplicate genomic loci with a significantly improved signal-to-noise ratioLarge array size affects Cas9 functionality[37]Organic dye-baseddCas 9dCas9-HaloTagEconomical, efficientAltered cell physiology[38]dCas 9sgRNA-MTS-MBDifferent combinations of fluorophore/bursting agent pairs and MB/MTS sequences can be flexibly selectedMBs are expensive[39]dCas 93sgMUC4-dual-MTS-MBDynamic imaging of non-repetitive genomic motifs with only three unique sgRNAsHigher sensitivityMBs are expensiveNo simultaneous visualization of multiple genomic motifs[40]Quantum dot (QD)-based imagingdCas 9dCas9-QDEnables tracking of individual virusesLarge size, difficult cell delivery[41]dCas 9SA-QDsEfficient labeling of internal viruses without altering the virus envelope and capsidLarge size, difficult cell delivery[42]RNA imagingFP-basedRcas9dCas9-FPImaging of RNALimit to imaging of low-abundance mRNA[43]FP-basedRCas9dCas9-EGFPImaging of RNALimit to imaging of low-abundance mRNA[44]FP-baseddCas9dCas9-EGFPAllow efficient imaging of low-abundance mRNALarge array size[45]Organic dye-baseddCas9sgRNA-Pepper530High fluorescence intensity and turn-on rateDetailed structure-activity of Pepper to be analyzed[46]FP-baseddCas13adCas13-EGFPImaging of RNANo current guidelines to design efficient gRNAs targeting an RNA of interest[47]FP-baseddCas13adCas13a-msfGFP-ZF-KRABOptimized S/N ratioImaging of intranuclear RNA is not possible[48]Organic dye-baseddCas13bCRISPR-TRA-tagModular design flexibilitySmaller and more accurate fluorescent signalsLarge array size[49]
Figure 2The CRISPR imaging system for genomic imaging. It relies on dCas9 or gRNA-bound fluorescent substances to visualize genomic loci in living cells. (**A**) dCas9 binds to a fluorescent protein (FP) [31]. (**B**) The supernova labeling (SunTag) system binds multiple fluorescent proteins [36]. (**C**) The SunTag system binds to light-induced nuclear output tags (LEXY) [37]. (**D**) RNA aptamers bind to molecular beacons MB [39]. (**E**) sgRNA carries two different molecular beacons (MBs). (**F**) dCas9 combined with DQ for imaging [40].
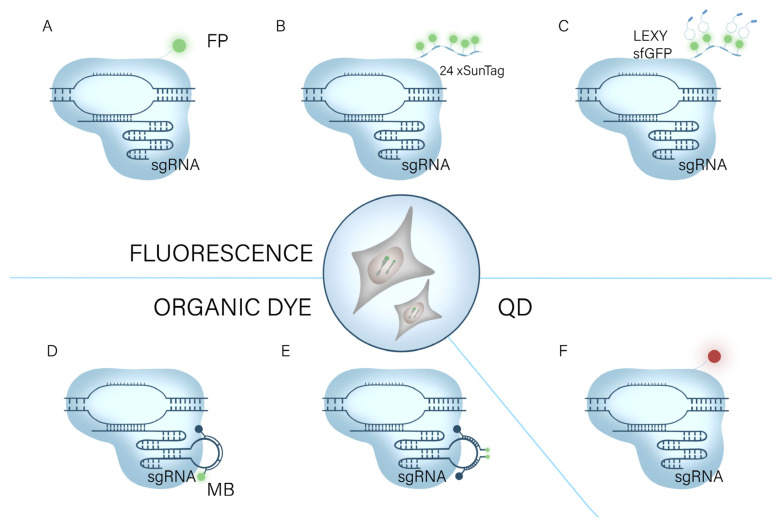


### 2.1. CRISPR/Cas for Imaging of Chromatin/Genomic Loci

#### 2.1.1. FP-Based CRISPR/dCas9 Systems

Numerous dCas9/gRNA systems have been developed to label genomic loci in living cells [50]. CRISPR/Cas9 system enables precise and programmed transcriptional activation and repression, modification, labeling, and visualization of local histones, as well as epigenetic re-regulation of DNA at genomic loci, and single-base genomic mutagenesis [50]. In a groundbreaking study, Chen et al., utilized EGFP to tag nucleic acid endonuclease structural domain HNH and RuvC-inactivated Pseudomonas dCas9 (SpCas9) for the precise and efficient labeling of non-repetitive sequences within the human genome. This innovative approach enabled researchers to observe the inherent conformation and dynamic properties of chromatin, shedding light on its natural behavior (Figure 2A) [31]. Chen marked a significant advancement in the field and made it possible to examine mitosis, telomere length, gene (chromosome) copy number, and gene dynamics in real time in living cells. However, the current limitation of scanning multiple genomic loci simultaneously in a single living cell poses a constraint on the comprehensive study of chromosomal dynamics [35]. To increase the use of CRISPR/Cas systems in functional genomic research, Chen et al. developed a multicolor imaging system based on the different DNA binding and PAM recognition specificity of dCas9 [34]. By combining SaCas9 and SpCas9, they were able to address the problem of double gene imaging at locus spacing of less than 300 kb. In addition, the CRISPRainbow system was enhanced by Ma et al., allowing the simultaneous detection of six loci and revealing considerable changes in the dynamic properties of numerous chromosomal loci [35].

However, the applications of each of these methods in low- or no-repeat DNA sequences are constrained by the relatively weak fluorescence signal and poor signal-to-noise ratio [11]. Signals in a variety of biological processes can be amplified by introducing many copies of the regulatory protein to the site of action [51]. Tanenbaum et al., leveraged this principle to pioneer the development of a repetitive peptide array known as SunTag [52]. By employing SunTag, they successfully recruited multiple copies of the transcriptional activation structural domain to the dCas9 protein, resulting in the creation of highly efficient synthetic transcription factors. Through this innovative approach, they demonstrated robust activation of endogenous gene expression and effectively redesigned cellular behavior (Figure 2B) [52]. Building upon this work, Ye et al., combined the SunTag system with the CRISPR/Cas9 system, leading to an amplification of the fluorescence signal and an improvement in the signal-to-noise ratio. This advancement enabled the labeling of low-repeat motifs and significantly broadened the range of applications for the CRISPR/Cas9-based imaging method [36]. Additionally, Hou et al., integrated the light-inducible nuclear export tag (LEXY) with the CRISPR SunTag system (Figure 2C). The non-targeted nuclear tagging module was light-controlled and transferred into the cytoplasm, considerably increasing the signal-to-noise ratio [37]. It is anticipated that CRISPR genome/chromatin imaging will have more applications in cell biology as it keeps improving.

#### 2.1.2. Organic Dye-Based CRISPR/dCas9 Systems

Fluorescent proteins are commonly used in CRISPR/Cas genomic imaging systems to mark genes. However, their limited photostability and fluorescence brightness pose challenges when labeling non-repetitive sequences or multiple loci, which may impact chromosomal dynamics. In contrast, organic dyes offer superior photostability and fluorescence brightness. The advantage of organic dyes lies in their ability to penetrate cells, but only a limited number of traditional fluorophores meet this criterion for effective intracellular labeling [53].

By replacing the N, N-dimethylamino substituents in tetramethylrhodamine with four-membered azetidine rings, Grimm et al., created fluorescent dyes for the first time that preserved both spectroscopic characteristics and cell permeability [54]. In a study by Deng et al., customized fluorescent dyes were chemically coupled to Halo Tag tags tied to dCas9 proteins to enable direct viewing of genomic loci in the nucleus [38]. Gene detection frequently employs molecular beacons based on the fluorescence aggregation quenching action of organic dyes [55]. Wu et al., created a novel fluorescent-dye-based CRISPR imaging system termed CRISPR/MB by combining dCas9 and molecular beacons (MB) (Figure 2D). Before being supplied to MB, dCas9 and sgRNA-MTS are co-expressed to target particular locations in cells. The target sites are subsequently illuminated by MTS through hybridization with MB. The CRISPR/MB hybridization method allows for a flexible selection of various fluorophore/burst pair and MB/MTS sequence combinations and can be a potential platform for research on chromatin activity [39]. Mao et al., modified the single-guide RNA (sgRNA) to bind two different molecular beacon (MB) target sequences. This innovative approach is anticipated to effectively minimize background signals resulting from incomplete bursting and non-specific protein binding of individual molecular beacons (Figure 2E) [40].

#### 2.1.3. Quantum-Dots-Based CRISPR/dCas9 Systems

Due to their high brightness, high photostability, and color-tunable emission characteristics, quantum dots are superior fluorescent markers to organic dyes and fluorescent proteins [56]. Therefore, the properties of quantum dots make them promising for live-cell imaging. For imaging HIV-1-integrated proviral DNA in latently infected cells, Ma et al., created a dual-color labeled CRISPR system with two different color quantum dots. The suggested approach has the potential to be both a potent tool for viral imaging and a multifunctional platform for imaging single genomic loci in living cells (Figure 2F) [41]. The technique can visualize the mechanism of viral infection by incorporating light-emitting quantum dots (QDs) into labeled viruses and tracking their movement. However, effectively labeling the internal viral components without altering the virus envelope and capsid remains a challenge, and existing strategies are not applicable to most viruses [42]. In an effort to address this issue, Yang et al., employed the CRISPR imaging technique and achieved successful tagging of nucleic acids from the pseudorabies virus (PRV) using quantum dots. By leveraging sgRNA-guided binding to viral nucleic acids, quantum dots were strategically incorporated with dCas and subsequently encapsulated into PRV during the virion assembly process [42]. The CRISPR/Cas-based quantum dot (QD) single-virus tracking (SVT) approach has enormous potential in virology since it can extract data on the underlying dynamics of viral infection to elucidate the mechanism of viral infection. Quantum dots are highly anticipated to surpass fluorescent proteins and dyes as a powerful tool for imaging chromatin in live cells. Their unique properties, such as superior brightness and photostability, hold great promise for advancing chromatin visualization. However, it is worth mentioning that the integration of CRISPR imaging and quantum dots has yet to be investigated in the realm of chromatin imaging. While this represents an exciting potential avenue, further research and development are required to fully leverage the capabilities of CRISPR imaging in conjunction with quantum dots for chromatin studies [57].

### 2.2. CRISPR/Cas for RNA Imaging

Since RNA has a wide range of information, structure, and enzymatic activity, the name “RNA World” has been used to describe it [58]. Using its diverse information, RNA imaging provides thorough information on the location, expression, degradation, storage, and regulation of numerous cellular RNA types [59]. RNA imaging is a crucial tool for understanding RNA biology, and the advancement of this technique has had a significant impact on both scientific and clinical biology research [60,61].

In 1997, fluorescent RNA was microinjected into living cells to start the process of developing RNA imaging of living cells [5]. The MS2/MCP system is the most widely used tool to study RNA dynamics [62], and the recent advancements in RNA-targeting platforms for Cas9 and Cas13 hold great promise for driving the progress of RNA-targeting technologies. Numerous genomic regions have been visualized using the CRISPR imaging method. However, RNA-specific live-cell imaging tools have only recently been created. In this section, we aim to provide an overview of recent advancements in CRISPR/Cas-based RNA imaging systems, offering valuable insights and potential directions for future research studies in this field.

#### 2.2.1. CRISPR/Cas9-Based RNA Imaging System

Genetic loci in living cells have been extensively clarified by using the CRISPR/Cas9 technology, which allows for real-time tracking and imaging of chromosome dynamics. It is simpler to utilize, more precisely targeted, and less disruptive to the physiological condition of living cells than other fluorescent molecules, and it can be expressed in fusion with a variety of fluorescent molecules [63]. Using a particularly designed PAMmer, Cas9 may target the binding or cleavage of RNA targets while avoiding binding to the matching DNA regions [64]. These lay the groundwork for RNA visualization investigations using dCas 9-based CRISPR imaging devices in living cells.

Nelles et al., used the CRISPR/Cas9 technique to investigate how RNA is visible in living cells for the first time [43]. This was the first instance of RNA by imaging using the CRISPR/Cas9 technique. Since U6-driven sgRNA is generally restricted to the nucleus, NLS-labeled dCas (RCas9) facilitates attachment to its sgRNA and engagement with the target mRNA before nuclear co-export with the target mRNA. They proved that the RCas9–mRNA interaction is stable enough to allow nuclear co-export. The recognition of RNA by RCas9 has little impact on regular RNA metabolism. RCas9 was able to recognize GAPDH, ACTB, CCNA2, and TFRC mRNAs in living cells, paving the way for future RNA tracking or imaging of individual endogenous RNA molecules with increased sensitivity using RCas9 targeting multiple transcript sites or employing multi-tagged dCas9 proteins (Figure 3A) [43]. In an extension of the RCas9 system, Batra et al., confirmed that Cas9 targeting RNA enables the imaging of microsatellite repeat amplified RNA associated with DM1, DM2, and CAG [44]. Individual mRNA labeling of high-abundance mRNAs becomes easier using CRISPR technology [45]. The inability to tag non-genetically encoded mRNA using imaging techniques has imposed limitations on our understanding of the functional distribution of endogenous low-abundance mRNAs within cells. To address this challenge, Sun et al., devised the CRISPR-Sunspot method, which leverages the SunTag signal amplification system. This innovative approach utilizes CRISPR/Cas9 to effectively detect and analyze low-abundance mRNAs, enabling a deeper exploration of their functional characteristics (Figure 3B) [45].

Fluorescent dye aptamers, capable of binding and activating fluorescent dyes, have been employed to visualize highly abundant cellular RNA species. Chen et al., introduced a novel aptamer, named Peppers, which offers a straightforward and dependable approach for imaging various RNA species within live cells. Peppers530-based CRISPR imaging, in contrast to fluorescent protein-based RNA CRISPR imaging, enables arbitrary changes in fluorescent labeling with straightforward washing and staining. This versatility is especially helpful when producing stable cell lines or transgenic animals (Figure 3C) [46].

#### 2.2.2. CRISPR/Cas13-Based RNA Imaging System

The CRISPR/dCas13 system, which was created recently, has been widely employed for intracellular RNA imaging and does not require genetic modification to obtain equivalent RNA labeling efficiency.

Yang et al., showed that the dCasS 13 system’s dPspCas13b and dPguCas13b are useful dCas13 imaging proteins that can label lncRNAs in mammalian cells. It was possible to image the RNAs of NEAT 1, SatIII, MUC 4, and GCN 4. (Figure 4A) [47]. dLwaCas13a has been used for β-actin mRNA imaging, and the team optimized the signal-to-noise ratio using a negative feedback regulation mechanism by Abudayyeh et al. (Figure 4D) [48]. Understanding RNA function requires having a visual representation of RNA kinetics. Although CRISPR/dCas13 systems have been developed, the effectiveness of dCas13 for RNA imaging is still limited. To thoroughly screen for homologous Cas13 for RNA imaging, eight previously unknown dCas13 proteins are available for RNA labeling, and two of them, dMisCas13b and dHgm4Cas13b, target endogenous MUC4 with equivalent or even greater efficiency [65].

Researchers have devised techniques for imaging RNA within active cells by integrating fluorescent aptamers with dCas13. Tang et al., introduced a highly efficient method for visualizing RNA dynamics in live cells, employing the CRISPR/dPspCas13b system in conjunction with fluorescent RNA aptamers. This innovative approach holds tremendous potential for advancing our understanding of RNA-related processes within living cellular environments (Figure 4E) [66]. Recent advancements have enabled precise and effective imaging and tracking of endogenous RNA by employing modified sgRNAs that are connected to fluorescent RNA aptamers, resulting in reduced background fluorescence. Furthermore, through washing and restaining procedures, it is now possible to effortlessly achieve color switching by altering the fluorescent dye analogs within living cells. These developments greatly enhance our ability to study RNA dynamics with improved precision and versatility. Tat peptide, a positively charged cell-penetrating peptide, has been recognized for its ability to enhance the cellular uptake of diverse molecules, including drugs and proteins [67]. By incorporating histidine and cysteine residues, an endosmotic Tat peptide has been developed as a non-viral vector for DNA transfection [68]. Building upon these advancements, Tang further combined it with the Tat peptide. Using the modified CRISPR/Cas RNA hairpin-binding protein fused with a Tat peptide array, modified RNA adaptors were recruited. Moreover, the modular design of CRISPR-TRAP-tag facilitates the replacement of sgRNA, RNA hairpin-binding protein, and aptamer to enhance imaging quality and live-cell affinity [49].

#### 2.2.3. RNA Imaging Based on other Cas Platforms

The application of labeled Cas9 and Cas13 in live-cell RNA tracking has brought about groundbreaking advancements, greatly broadening the range of tools available in this domain. However, both the Cas9/Cas13 platform and the commonly used MS2-MCP technology have faced challenges in mitigating the problem of excessive background noise, which has restricted their effectiveness in certain situations. Although exogenous tandem RBS with fairly high copy numbers (typically > 24 copies > 1000 bp) can be used to label target RNA [69], genetic manipulation can be a complex process and may inadvertently impact the structure and localization of target RNAs. However, Han et al., introduced an innovative approach to address these challenges. They developed a Cas6-based fluorescent complementation platform (Cas6FC), which offers a novel solution for detecting target RNAs with minimal background noise. This advancement provides a promising avenue to enhance the signal-to-noise ratio of CRISPR systems [70].

Most RNA-targeting tools showed significant cytotoxicity [71], but in 2021, researchers from the McGovern Institute at MIT found that Cas7-11 did not affect cell viability, implying that Cas7-11 can track RNA without harming cells [72]. Cas7-11, in addition to being a valuable research tool, also provides a good platform for the development of RNA imaging. Despite being a relatively new development in the field of RNA imaging, CRISPR/Cas systems have shown great potential in this area. Previous studies have highlighted their remarkable utility for RNA targeting. Leveraging the capabilities of CRISPR/Cas systems, there is a promising opportunity to develop sensors that can accurately detect and identify gene expression patterns associated with specific diseases [73].

### 2.3. CRISPR/Cas for Protein Imaging

Proteins are the direct performers of life activities and are involved in almost all life processes, including genetics, development, reproduction, and the metabolism of substances and energy [74]. Protein research cuts across all areas of scientific research, such as bio-diagnostics and bio-therapeutics, and it has huge research potential [75]. Uncovering the specific functional mechanisms of thousands of proteins in organisms is the core of protein research and one of the most challenging areas of life science research in the post-genomic era. Proteomic research aims to elucidate the functions of genes at the protein level, which is of great importance for exploring the mysteries of life. Among them, protein imaging technology has become one of the most important tools to study proteins. This technique allows the labeling of different proteins using fluorescent dyes, fusion proteins, etc., and visualization in cells. Currently, the CRISPR/Cas system is commonly used in protein imaging, which allows efficient and precise editing of target genes, resulting in rapid knockouts or knock-in of target genes, most often with the Cas12a and Cas9 proteins.

#### 2.3.1. Fluorescent Protein Labeling

Mikuni et al., were pioneers in the development of a novel technique known as CRISPR/Cas9-mediated homology-directed repair (SLENDR). This innovative approach allows for the labeling of endogenous proteins within individual cells. This approach allows for in vivo protein imaging specifically in the mammalian brain, opening up new avenues for studying protein dynamics and function in their native cellular context [76]. Epitope tagging plays a crucial role in live cell protein imaging applications. Daichi Kamiyama et al., presented a method by harnessing self-complementing split fluorescent proteins as epitope tags for live cell protein labeling. Through the use of CRISPR-mediated homology-directed repair, they effectively incorporated these tags into endogenous genomic loci. Their research highlighted the remarkable versatility of FP11-tags as a potent tool for protein labeling and imaging, opening up new avenues for studying protein dynamics in living cells [77]. In their study, GFP11x7 was well tolerated for a variety of proteins, including IFT20 and Cas9. However, GFP-based approaches showed reduced detection ingeniously. To circumvent this limitation, Schwinn MK and collaborators made use of CRISPR-mediated bioluminescent peptide (HiBiT) to label endogenous proteins in Hela cells, integrating HiBiT peptide tag was successfully incorporated into the genetic code, thus allowing for real-time quantitative protein detection [78]. Jelmer Willems et al., developed ORANGE to epitope-tag neuronal endogenous proteins using the genome-editing toolbox of CRISPR/Cas9. Multiple knock-in mechanisms were then created in neurons to mediate multiple imaging of endogenous proteins [79].

In the course of the application of CRISPR/Cas9 in protein imaging, other Cas proteins have been studied accordingly. Baldering et al., introduced CRISPR/Cas12a for simple and efficient labeling of endogenous proteins with photoactivated proteins [80]. They used the CRISPR/Cas12a system as a whole to introduce a new genome engineering method (called polymerase chain reaction (PCR) labeling) to label naturally occurring proteins, which renders it an intriguing labeling tool. Baldering et al., demonstrated that CRISPR/Cas12a endogenous tagging can be combined with quantitative super-resolution imaging. Chen et al., combined dLwaCas13a with HA Tag to develop CRISPR-based RNA interaction proteomics (CBRIP), which used dCas13 to track specific RNA and then used UV light to cross-link rbp and RNA [81].

#### 2.3.2. Near-Infrared Imaging

NIR-I (NIR-I, 700–900 nm) and NIR-II (NIRII, 1000–1700 nm) are the two main NIR wavelength ranges used in near-infrared (NIR) fluorescence imaging [82]. Biological tissues exhibit lower light absorption and scattering in the near-infrared (NIR) band compared to the visible spectrum. This characteristic of NIR light presents significant advantages in the context of providing physiological and pathological information while minimizing harm to tissues and reducing interference from background fluorescence. NIR fluorescence imaging offers numerous benefits, including enhanced tissue penetration, reduced background noise, improved signal-to-noise ratio, and increased sensitivity. These advantages contribute to more effective and reliable imaging, allowing for a better understanding of biological processes in vivo [83]. Butkevich and his colleagues combined NIR technology with CRISPR/Cas9 for the exogenous tagging of significant proteins. This method allowed for the stable production of the fusion protein, providing reliable and other intercellular reproducible two-color nanotechnology cinematography that can be dependable and repeatable in cells [84]. However, for photostability, the quantum yield of each NIR probe is still much lower than that of the corresponding type of probe emit-ted in the visible range. And there is still a greater space to further improve their quantum yields [83]. Imaging technologies based on NIR and CRISPR/Cas9 have not been thoroughly investigated. NIR fluorescence imaging exhibits deep penetration and great resolution in biological tissues. NIR fluorophores and CRISPR/Cas9 have a lot of innovation and potential for multicolor imaging as a result of their combined application.

### 2.4. Deficiencies and Shortcomings of CRISPR/Cas for Live-Cell Imaging

Improving the signal-to-noise ratio. Improving the proportion of signal to noise in CRISPR imaging is a major challenge. To increase the signal value, researchers have amplified the signal by recruiting more fluorophores at a single site [36,37,45], and to increase the fluorescence intensity, researchers have replaced them with higher-quality fluorescent markers [39,40]. CRISPR imaging systems based on fluorescent dyes have been studied accordingly. However, fluorescent dyes have problems such as high price and cytotoxicity, which limit their application. The emergence of quantum dots (DQs) with high brightness, good photostability, and color-tunable emission properties [41,42,56] has provided new ideas for the development of CRISPR imaging technology.

Targeting specificity issues is a challenge. In RNA imaging, due to the rapid synthesis and metabolism of RNA in vivo, intracellular RNA imaging using Cas protein and sgRNA-mediated imaging is far more difficult than DNA, and targeting and tracking each type of RNA is more difficult [85,86]. The advent of RCas9 has made CRISPR-based RNA imaging possible, and endogenous coding RNA visualization has been achieved, but non-coding RNA imaging needs further study. In addition to RCas9, dCas13 with higher targeting precision is also being gradually employed for gene imaging. The optimized dCas13 system is user-friendly, does not require genetic manipulation, and achieves higher RNA labeling efficiency [47]. By combining pre-crRNA processing null dMisCas13b (ddMisCas13b) with RNA aptamer-fused gRNAs, Yang et al., further developed the CRISPR palette system, which has visualized RNAs in triple colors in living cells and improved the target specificity of CRISPR/Cas13 in RNA imaging [87]. The current difficulties in RNA imaging are mainly focused on the targeting of RNAs, and efficient gRNA design guidelines for RNAs are yet to be developed [65].

Various components/biomolecules need to be imaged simultaneously. The majority of recent research has concentrated on single-target imaging in living cells. We can better grasp the linkages between different biological macromolecules, such as DNA, RNA, proteins, etc., by simultaneously imaging many molecules. To gain fresh insights into the coordinated control of many processes involved in molecular biology’s basic dogma, further technologies for the simultaneous imaging of multiple components and biomolecules are required [32].

## 3. Research Progress of CRISPR/Cas for Bioanalysis

Molecular diagnostic technologies have become a new generation of widely recognized diagnostic techniques worldwide [88]. CRISPR/Cas technology can be combined with biosensors and bio-detection methods for molecular diagnostics [89,90,91,92,93,94]. Due to its high specificity and sensitivity, programmability, and device independence, CRISPR-based detection has gained widespread attention as a sensor [92,95,96,97,98]. Currently, nucleic-acid-based detection is the most extensively studied approach. However, there is a growing interest in expanding the scope of detection beyond nucleic acids to include non-nucleic acid targets, such as small molecules and proteins. Further efforts are underway to explore and develop detection strategies for these targets [99,100,101]. This section reviews the research progress of CRISPR systems in biosensing, summarizes representative biosensors (Table 2), and discusses the challenges of CRISPR-based biosensing.

### 3.1. Nucleic Acid Analysis

DNA sequences and RNA transcripts play crucial roles in various biological events and can serve as biomarkers for biological research and clinical diagnosis. Electrophoresis and blotting are the most widely used analytical techniques for analyzing nucleic acids, but their sensitivity is limited to sub-microgram ranges. CRISPR/Cas is a highly effective method for nucleic acid detection, based on specific enzyme systems to recognize, locate, and cleave target DNA or RNA sequences. Currently, numerous nucleic acid sensing systems have been developed using various CRISPR/Cas systems, showcasing their potential for the development of highly sensitive, super-resolution, cost-effective, and time-saving detection methods [114].

#### 3.1.1. Cas9-Based Nucleic Acid Detection System

The combination of the CRISPR/dCas9 system with fluorescent dyes has been extensively utilized for live-cell imaging purposes [34,35,36,115]. Unlike imaging, the requirements for biological analysis have higher accuracy, and the off-target issue of CRISPR/dCas9 may lead to false-positive detection results [116]. Zhang et al., introduced an innovative in vitro DNA detection system that involved the fusion of a pair of dCas9 proteins with split fragments of luciferase. This system exhibited luminescence activation exclusively when the reporter genes co-localized with the 44 bp target sequence specified by sgRNAs. The system’s output depended entirely on sequence and spatial constraints, yielding a remarkably specific reporting system [117].

The CRISPR systems can be utilized for nucleic acid analysis via fluorescent dye binding and by amplifying nucleic acid signals through the Cas9 protein’s specific cleavage activity. Pardee et al., incorporated the Cas9 effector with isothermal RNA amplification technology NASBA [118] to develop the first CRISPR-based nucleic acid detection system, and it was able to differentiate between Zika genotypes with single-base resolution within three hours (Figure 5A) [92]. Subsequent studies have combined CRISPR/Cas9 with various isothermal amplification methods, such as exponential amplification reaction (EXPAR) [119] and rolling circle amplification (RCA) [120], to expedite detection times and improve sensitivity and specificity. Huang et al., developed the CAS-EXPAR detection system, which uses the EXPAR amplification system and can complete long-chain RNA detection within one hour with a detection of limit of 0.82 aM. Meanwhile, such system was successfully employed in detecting DNA methylation and single-cell proliferation of Listeria monocytogenes total RNA (Figure 5B) [102]. RCA is another highly specific isothermal amplification method. Wang et al., combined RCA with the Cas9 effector to establish a nucleic acid detection platform called RACE, which can detect miRNAs at a single-base resolution simultaneously with a detection limit of 9 fM [103].

Nucleic acid amplification technology can amplify nucleic acid signals and enhance the sensitivity of detection. However, nucleic acid analysis systems based on amplification technology undoubtedly increase the complexity of testing time and operation. With the development of electrochemistry, the emergence of E-DNA sensors provides a possibility for non-amplified nucleic acid detection [121]. Xu et al., combined the CRISPR/Cas9 system with E-DNA sensors to achieve unprecedented sensitivity and accuracy in non-amplified conditions, realizing the detection of ssDNA viruses and reaching the PM detection limit (Figure 5C) [104].

#### 3.1.2. Cas12-Based Nucleic Acid Detection System

After binding to the target DNA, Cas12a/b releases non-specific ssDNA cleavage activity. This target-induced trans-cleavage activity has been utilized for the detection of DNA molecules [24]. Li et al., first developed the HOLMES system using this feature for human genotyping, and it can detect DNA and RNA viruses with a 1 aM detection limit within 1 h (Figure 6A) [90]. Chen et al., developed a detection platform called DETECTR, which allows for the quick and precise detection of human papillomavirus (HPV) at micro-molar levels (Figure 6B) [24]. Compared to Cas12a, Cas12b has a wider temperature range for trans-cleavage activity. Building upon the HOLMES platform, Wang et al., successfully integrated LAMP with Cas12b trans-cleavage activity, creating a streamlined one-step system. This integration simplifies the detection process, reduces time, and minimizes the instrument requirements, thereby enhancing the overall efficiency and practicality of the system (Figure 6C) [105]. To further enhance detection sensitivity, Zhang developed a dual amplification sensing method utilizing terminal deoxynucleotidyl transferase (TdT) and CRISPR/Cas12a (Figure 6D). The dual amplification steps led to a synergistic signal amplification effect caused by PNK and can detect PNK activity in cell lysate and screen inhibitors with high selectivity and sensitivity. This method was successfully applied to PNK inhibitor screening and PNK activity detection at the single-cell level [98].

The ssDNA reporter gene in HOLMES or DETECTR can be labeled with paired fluorescence/quenching agents for fluorescence detection, or combined with a colorimetric strategy in LFA for visual inspection. Wang et al., proposed a highly sensitive visual nucleic acid detection method, which utilizes fast PCR amplification in a portable insulated cup and a body-heat-triggered Cas12a cleavage reaction [97]. With the aid of mini-UV touch, the fluorescence results can be easily identified by the naked eye. This greatly simplified the operation, completely avoided amplicon contamination, and realized a high-performance and minimally equipped nucleic acid detection platform that could be deployed on-site. Kyeonghye et al., have developed a novel hybrid Cas protein approach for detecting nucleic acids (DNA and RNA) for multiplexed detection. The CRISPR/hybrid Cas system recognizes nucleic acids simultaneously, and dual detection of pathogenic viruses can be performed in a single tube [122]. The hybrid Cas protein has the potential to be used in molecular diagnostic methods in infectious diseases and tissue and liquid biopsy, and other nucleic acid biomarker detection [106]. Weng et al., conducted a study to demonstrate the intrinsic signal amplification and ultra-sensitivity of graphene field-effect transistors (gFETs) achieved through the multiple trans-cleavage activities of CRISPR/Cas12a. This breakthrough enabled the detection of ssDNA and dsDNA targets without the need for target pre-amplification. The CRISPR Cas12a-gFET system achieved reliable and highly sensitive detection at attomolar (aM) levels for human papillomavirus (HPV) and *E. coli* plasmid [94].

#### 3.1.3. Cas13-Based Nucleic Acid Detection System

Cas13 can perform RNA-guided RNA targeting by cleaving ssRNA. The discovery of collateral cleavage activity has also simplified the detection methods using CRISPR systems. Cas13a was found to have RNA-targeting collateral cleavage activity triggered by target RNA in 2016 [123]. Building on this activity, the Gootenberg team developed the first comprehensive and applicable nucleic acid detection system based on CRISPR/Cas13a, called SHERLOCK, which achieved aM sensitivity for single-base resolution detection of targets (Figure 7A) [30]. Furthermore, Myhrvold et al., discovered that the combination of HUDSON and SHERLOCK enabled direct virus detection from bodily fluids without instrumentation, achieving DENV detection (Figure 7B) from patient samples in less than two hours and demonstrating the potential of instrument-free detection of clinically relevant viral single-nucleotide polymorphisms [91].

The emergence of the SHERLOCK system has brought a new wave of changes to nucleic acid detection technology. To expand SHERLOCK’s sample multiplexing capabilities, the Gootenberg team utilized multiple SHERLOCKs with PsmCas13b and LwaCas13a to achieve multi-target detection of ZIKV and DENV RNA dilutions and human saliva samples for allele-specific genotyping with orthogonal primer preference and sample multiplexing. These advances allow the detection of many targets on a large scale and are more cost-effective (Figure 7C) [89]. In 2020, the Wilson team introduced an orthogonal pipeline called CREST. This innovative approach combines traditional and reliable biochemical methods, such as PCR, with low-cost instrumentation, without compromising detection sensitivity. CREST utilizes simple fluorescence visualization tools, enabling an intuitive display of results (Figure 7D) [93].

#### 3.1.4. Other Cas-Based Nucleic Acid Detection Systems

The Cas12f family is characterized by its compact size, approximately 400–700 amino acids, which is half the size of other recognized class II CRISPR/Cas systems. These nucleases possess a distinct capability to bind and cleave target single-stranded DNA (ssDNA) with high specificity, guided by tracrRNA and crRNA (or sgRNA), without the need for a PAM site restriction. Cas12fa adopts a V-shaped conformation and can interact with target nucleic acids, activating its ssDNA trans-cleavage activity [124]. This unique property facilitates the molecular detection of target nucleic acids [107]. The potential of Cas12f for biosensing analysis is still under development. Cas12f’s target-dependent non-specific DNase activity is used as a DNA detection platform for high-fidelity detection systems (Cas12f-DETECTR) [107]. In comparison to Cas12a-DETECTR, Cas12f-DETECTR demonstrates enhanced specificity and activity, enabling high-fidelity detection of DNA single nucleotide polymorphisms (SNPs) [125]. Zhou et al., developed a fluorescent biosensor known as HARRY (High-Sensitivity Aptamer-Regulated Cas12f R-Ring for Bioanalysis) [126]. In the absence of a target, single-stranded DNA (ssDNA) activates Cas12fa, resulting in the trans-cleavage of a fluorescent reporter gene and subsequent fluorescence enhancement. However, in the presence of a target, the aptamer interacts with the ssDNA, forming an ssDNA-target assembly. This assembly inhibits the activation of Cas12fa, preventing the cleavage of the fluorescent reporter gene. However, compared to other CRISPR/Cas systems, Cas12f has a far less developed body of research and offers promise for biosensing analyses.

### 3.2. Protein Analysis

Based on the specific binding activity and efficient trans-cleavage capability of CRISPR/Cas systems [1], combined with electrochemical [110] and fluorescence sensing technologies [127], it has successfully achieved high-specificity and high-sensitivity detection of various target molecules, and the detection objects have been expanded from the initial nucleic acids to metal ions, proteins, bacteria, small molecules, etc. In protein analysis, CRISPR/Cas systems are often used as signal amplifiers because non-nucleic acid target molecules cannot be directly recognized, and often require the incorporation of recognition elements, such as functional nucleic acids (deoxyribozymes and aptamers), antibodies, and bacterial allosteric transcription factors.

In the field of fluorescence sensing technology, to overcome the high background signal generated by exonuclease-assisted cyclic signal amplification, chain displacement amplification [128,129], cross-chain reaction [130], rolling circle amplification [131], etc., Du et al., catalyzed substrate elongation by Cascade Terminal Deoxynucleotidyl Transferase (TDT) [132] and CRISPR/Cas12a catalyzed short-strand DNA probe cleavage (Figure 8A) [108]. The Cas12a effector can be activated and cleave the single-stranded DNA probe, thereby generating a fluorescent signal. This biosensor was used for ultrasensitive detection of UDG and T4 PNK with detection limits of 5 × 10^−6^ U/mL and 1 × 10^−4^ U/mL, respectively [133]. Chen et al., further developed a mechanism (Figure 8B) [134] that exploits a CRISPR/Cas13-based signal output amplification strategy, employing dual antibodies to capture proteins. The output fluorescence signal was amplified by the transcription of CRISPR/Cas13a side-cleavage activity by T7 RNA polymerase. Fluorescence microscopy successfully detected inflammatory factors, such as human interleukin, tumor markers, and human VEGF, with a sensitivity that was at least 102-fold higher than conventional ELISA methods [109]. Nevertheless, the existing fluorescence acquisition modes rely on traditional Stokes emission (visible excitation), which limits their capability to effectively analyze complex samples due to insufficient power [135]. Li et al., found a promising up-conversion nanoparticle (UCNP) luminescent material to overcome this weakness [135,136,137]. In addition to its excellent optical properties, UCNP has a very narrow half-width (<20 nm) and good photobleaching resistance. The conversion of near-infrared (NIR) photons to visible photons can significantly eliminate biological background interference [138]. Taking advantage of this property, Li et al., first designed a multiplex light trapping strategy to realize two-component detection by introducing holographic OT and upconversion luminescence coding and proposed a novel CRISPR/Cas12a biosensor (Figure 8C) [139], which combined energy-limited enhanced upconversion LRET with biomimetic interface-assisted luminescence enhancement, and applied it to functional DNA regulatory transduction of non-nucleic acid targets.

Unlike conventional optical methods, the utilization of an electrochemical DNA (E-DNA) sensor based on this strategy allows for the detection of target DNA without the need for complex and costly optical components, light sources, or photodetectors. This electrochemical sensing approach offers a more streamlined and cost-effective solution for sensitive DNA detection [121]. Dai et al., introduced the pioneering work on the development of an electrochemical biosensor (E-CRISPR) based on the CRISPR/Cas12a (Cas12a) system, leveraging the exceptional target recognition capability of protein aptamers [110]. The E-CRISPR platform was utilized to quantify the transforming growth factor beta 1 (TGF-β1) protein in a clinical sample. In the presence of the protein target, ECRISPR exhibited reduced binding of aptamers, resulting in a higher electrochemical signal generated by the methylene blue from the ssDNA reporter. Conversely, in the absence of the protein target, the activation of trans-cleavage activity by target recognition led to a lower electrochemical signal. This mechanism allowed for sensitive and specific protein detection using the ECRISPR platform. The experimental detection limit was determined to be 0.2 nM.

In general, the development of CRISPR/Cas technology has brought many new methods to study the structure and function of proteins. These methods take advantage of the high target specificity and flexibility of CRISPR/Cas technology to make protein analysis easier, faster, and more accurate.

### 3.3. Small Molecule Analysis

Small-molecule compounds exist widely in cells, such as ATP, AMP, amino acids, monosaccharides, and so on [140]. The small molecule is an important component of a macromolecule. They have a significant impact on human health, the environment, and food safety [141]. As a next-generation molecular diagnostic tool, CRISPR/Cas enables the precise detection of a variety of small molecules, particularly utilizing side-cleavage activity found in Cas12 or Cas13, allowing the development of a variety of supersensitive detection tools. It can be combined with aptamers [100] and so on to establish new biosensors or new strategies to achieve more efficient, simple, fast, and accurate detection, and to achieve the simultaneous detection of a variety of small molecules, broadening the application of CRISPR/Cas systems in the medical field.

Adenosine triphosphate (ATP) plays an important role in providing energy for cellular activities, participating in various biochemical reactions and cellular metabolism. Its presence serves as a marker for assessing cell proliferation, detecting microbial infections, and analyzing environmental conditions, due to its involvement in a wide range of biological processes [100]. ATP concentrations are tightly regulated under normal conditions. However, the majority of biosensors utilizing the CRISPR/Cas system have predominantly targeted nucleic acids, with limited exploration of small-molecule CRISPR technologies. Therefore, there is a pressing need to develop additional CRISPR/Cas biosensors that can expand the range of targets beyond nucleic acids, thereby broadening the potential applications of this technology [99].

Peng et al., made a groundbreaking contribution by integrating the CRISPR/Cas12a system with an adapter, resulting in the development of an adapter-mediated fluorescent biosensor. This innovative biosensor enables highly sensitive detection of adenosine triphosphate (ATP) through the utilization of fluorescence signals. Their innovative approach enabled the detection of ATP at an impressively low concentration of 400 nM [100]. On this basis, Wang et al., added the Strand Displacement Amplification (SDA) amplification step [101] and developed a four-stage ATP measurement signal amplification method. The amplified template of SDA serves a dual purpose: it facilitates efficient cyclic amplification of the target and effectively mitigates self-extension and non-specific amplification of transient templating. As a result, sensitivity is greatly enhanced, and the detection limit is significantly reduced. Niu et al., made a noteworthy breakthrough in understanding CRISPR/Cas12a, uncovering that a concentrated amount of activator (ssDNA or dsDNA bound to crRNA) can effectively suppress the trans-cleavage activity of Cas12a. Expanding on this unique feature, they devised a versatile diagnostic method for small molecules called Molecular Radar (Random Molecular Aptamer-Dependent CRISPR-Assist Reporter). This approach enables swift and remarkably sensitive detection while maintaining exceptional specificity. The Molecular Radar platform achieves an impressive detection limit as low as nM, highlighting its remarkable sensitivity [99].

In the electrochemical sensor-based CRISPR ATP detection, Xu et al., provide an efficient ECL biosensing system based on CRISPR/Cas12a for the determination of ATP [142]. The CRISPR/Cas12a system can convert the recognition of target ATP into detectable electrochemiluminescence (ECL) signals on bipolar electrodes. This system harnesses the remarkable potential of CRISPR/Cas12a in accurately recognizing and quantifying ATP targets, facilitated by the discovery of its cleavage capability towards single-stranded DNA (ssDNA). The sensor described in this study utilizes the transformation of ATP recognition into cleavage activity on Fc-ssDNA, thereby regulating electrochemiluminescence (ECL) signaling. This approach enabled sensitive quantitative analysis of ATP, with a detection limit of 0.48 nM. Niu et al., achieved higher sensitivity by changing the F-Q probe to an on-electrode modified MB-ssDNA reporter for application in electrochemical biosensors, expanding the field of application [99]. Furthermore, in conjunction with ECL sensors, Zhao et al., employed Fe_3_O_4_ magnetic nanoparticles (MNPs) for the modification of single-stranded DNA (ssDNA) on their surface. The utilization of Fe_3_O_4_ MNPs was driven by their distinct magnetic properties, non-toxicity, chemical stability, and biocompatibility. This approach facilitated the efficient introduction of ssDNA and the subsequent removal of unreacted ssDNA, thereby preventing unintended trans-cleavage of ssDNA. The developed sensor achieved a detection limit of 1.9 nM [111].

Li et al., presented a novel approach called bivalent aptamer-assisted CRISPR/Cas12a-mediated transversal flow assay (BA-CASLFA) to address the limitations of CRISPR/Cas12a sensors [112]. The innovative platform introduced here utilizes bivalent aptamers, enabling fast binding to ATP and eliminating the requirement for an enzyme inactivation step. By harnessing the efficient nucleic acid lateral flow assay (LFA) signal output system, this platform achieves ATP detection in just 26 min, with a visible “TURN ON” signal detectable to the naked eye. Additionally, the BA-CASLFA platform demonstrates versatility and can be adapted for the detection of various other small molecules.

Wang et al., introduced a luminescent nanoplatform, integrating CRISPR/Cas12a with zeolite imidazolate framework (ZIF-90)@Ag_3_AuS_2_@Fe_3_O_4_ nanocomposites, to enable multi-channel detection [113]. The newly developed platform allows for the simultaneous quantification of ADP and ATP, providing absolute concentration measurements of the nucleotides instead of relying on the ADP/ATP ratio. By leveraging highly specific binding and robust signal amplification, the detection limits for ADP and ATP reached as low as 0.022 nM and 0.079 nM, respectively, within a 30-min time frame.

### 3.4. Flaws and Shortcomings of CRISPR/Cas Systems for Bioanalysis

The versatile and adaptable characteristics of CRISPR/Cas technology enable its seamless integration with various signal detection platforms, providing enhanced flexibility and functionality for diverse applications. Moreover, sgRNA is more specific in identifying target complementary sequences compared to primer-based PCR assays [143]. Numerous CRISPR/Cas-based bioanalytical studies are currently in the laboratory stage, facing common challenges such as high detection costs, prolonged incubation times, and the utilization of complex detection platforms [144,145,146,147,148]. Consequently, there is an urgent need to advance the development of diagnostic devices that are more rapid, sensitive, intelligent, and portable, enabling practical and accessible implementation of CRISPR/Cas-based bioanalysis methods. For example, the low-cost and pocket-sized fluorescence detector developed by Katzmeier et al. [149] provides a low-cost, promising method for immediate nucleic acid detection (POCT) based on CRISPR/Cas13 trans-activity. At the same time, CRISPR/Cas technology has the same problems of PAM sequence restriction, off-target activity, insufficient insertion–deletion, or inefficient homology-directed repair (HDR) in the field of bioanalysis [150] to be solved. With further improvements in these directions, the potential application areas of CRISPR/Cas technology appear to be boundless. The utilization of CRISPR/Cas systems in bioanalysis holds the promise of delivering unparalleled quality and desirable benefits to humanity, paving the way for groundbreaking advancements in various fields.

## 4. Outlook

CRISPR/Cas technology provides a versatile platform for targeting specific nucleic acids, enabling their precise imaging and analysis [151]. The continuous advancements in CRISPR/Cas technology have significantly contributed to the accurate visualization of chromatin structure, genomic loci, RNA molecules, and proteins within cells. The exceptional sensitivity and rapid detection capability of the CRISPR/Cas system have positioned it as a crucial tool in clinical applications, particularly for nucleic acid, protein, and small molecule detection. These innovative initiatives underscore the immense potential of CRISPR/Cas technology in bioanalysis and live-cell imaging, providing invaluable insights into cellular processes and molecular interactions. Although Cas9, Cas12a, Cas13, and Cas12f have received significant attention for their utility in genome editing and nucleic acid detection, ongoing research aims to discover and engineer evolved CRISPR/Cas systems with improved assay sensitivity and reliability. These advancements will unlock new possibilities for broader applications in diverse fields, including biotechnology, chemical sciences, clinical medicine, and materials science. The continuous progress in CRISPR-based imaging and bioanalysis holds great promise for transformative advancements in these disciplines.

## Figures and Tables

**Figure 1 ijms-24-13447-f001:**
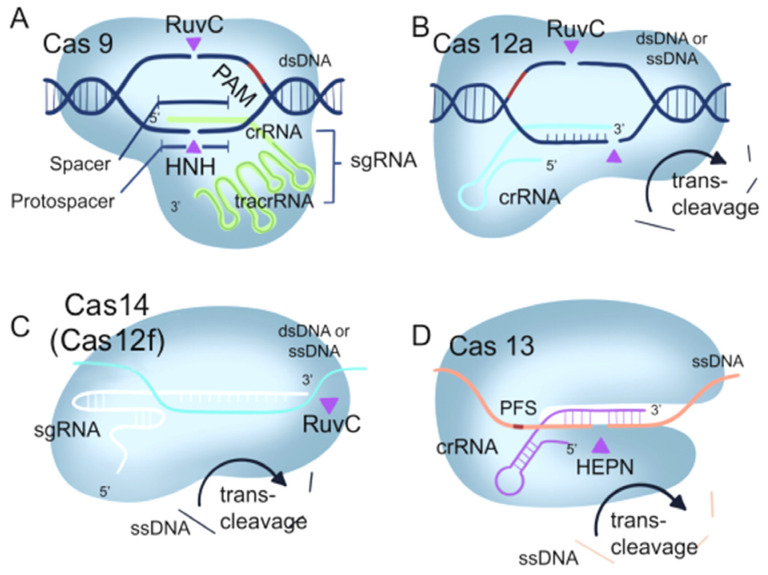
Fundamental components of four CRISPR/Cas systems. (**A**) CRISPR/Cas9. (**B**) CRISPR/Cas12a. (**C**) CRISPR/Cas14(Cas12f). (**D**) CRISPR/Cas13a. Purple arrowheads indicate cis-cleavage sites, and black arrows indicate trans-cleavag.

**Figure 3 ijms-24-13447-f003:**
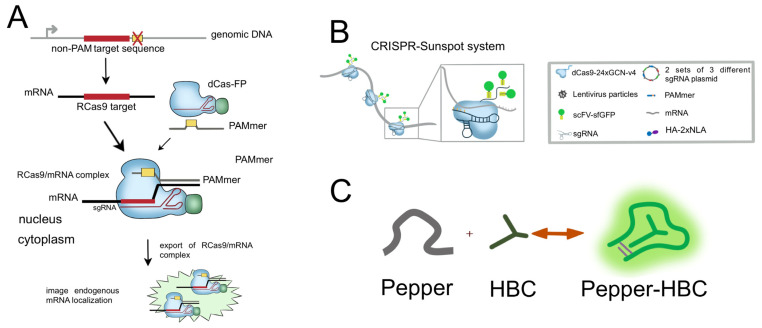
Using RCas9 to target mRNA in living cells. (**A**) Schematic diagram of RNA imaging system using CRISPR/dCas9 [43]. The cross symbol represents sequences that do not contain the PAM sequence. (**B**) Schematic diagram of the CRISPR-Sunspot imaging strategy, depicting the utilization of three target sites within a single mRNA for visualization purposes [45]. (**C**) Schematic representation of the Pepper530 complex [46].

**Figure 4 ijms-24-13447-f004:**
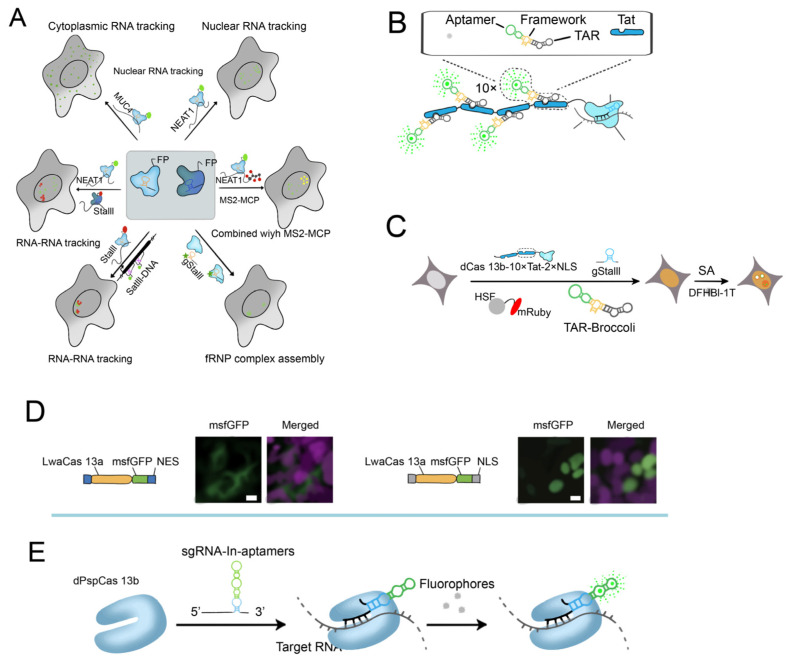
Targeting mRNA in living cells. (**A**) Strategy for imaging NEAT 1, SatIII, MUC 4, and GCN 4 RNAs in live cells using the CRISPR/dCas13 system [47]. (**B**) Development of RNA imaging strategy by a combination of Tat peptide with fluorescent RNA aptamers (TRAP-tag) [49]. (**C**) Schematic diagram of applying CRISPR-TRAP-tag to track SatIII lncRNA under SA treatment [49]. (**D**) Imaging demonstrating the localization and expression patterns of each mammalian construct [48]. Scale bars, 10 μm. (**E**) Schematic of CRISPR/dCas13 system binding with sgRNA containing [66].

**Figure 5 ijms-24-13447-f005:**
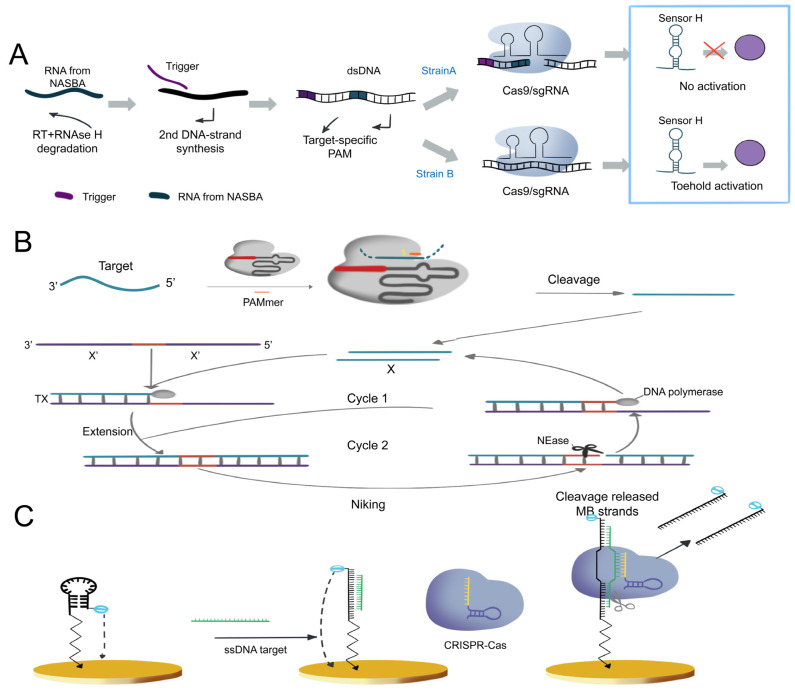
Nucleic acid detection based on CRISPR/Cas9 system. (**A**) Nucleic acid detection based on NASBACC [92]. (**B**) CAS-EXPAR nucleic acid analysis system [102]. (**C**) CRISPR/Cas9 nucleic acid analysis system in conjunction with E-DNA sensors [104].

**Figure 6 ijms-24-13447-f006:**
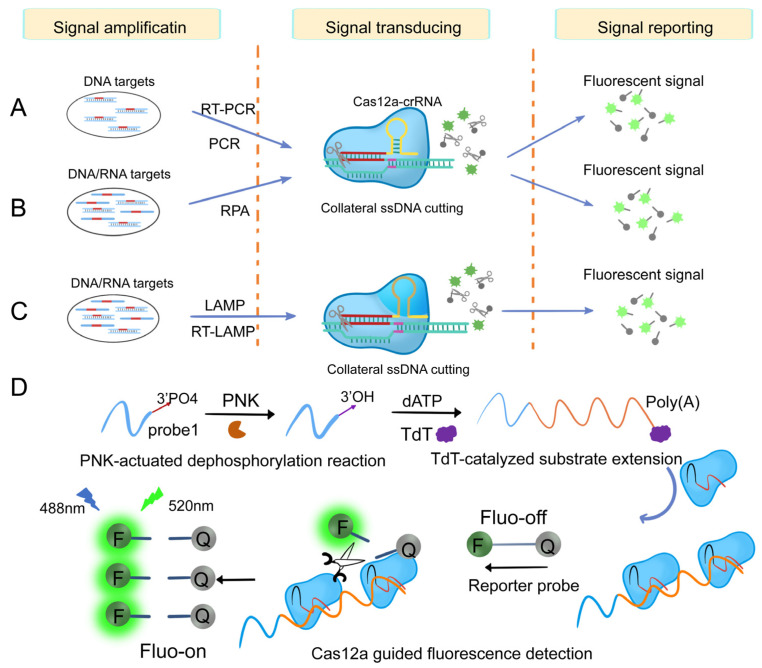
Nucleic acid analysis system based on Cas12. (**A**) Schematic of direct nucleic detection acid using HOLMES system [90]. (**B**) Schematic of direct nucleic acid using DETECTR system [24]. (**C**) Schematic of direct nucleic acid using HOLMESv2 system [105]. (**D**) Double-amplification sensing strategy based on terminal deoxynucleotide transferase (TdT) and CRISPR/Cas12a [98]. Scissors represent nicking endonuclease (Cas12a) cleaving the reporters.

**Figure 7 ijms-24-13447-f007:**
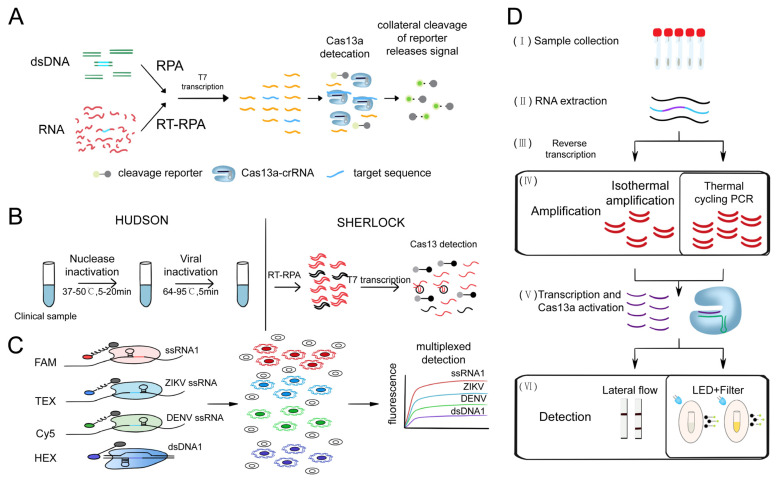
Schematic diagram of nucleic acid detection system based on Cas13. (**A**) Schematic of ZIKV RNA detection by SHERLOCK [30]. (**B**) Schematic of direct viral detection using HUDSON and SHERLOCK [91]. (**C**) Overview of Cas13-based CREST modifications [89]. (**D**) Schematic of complementary Cas13 and Cas12a enzymes in 4-channel in-sample multiplexing [93].

**Figure 8 ijms-24-13447-f008:**
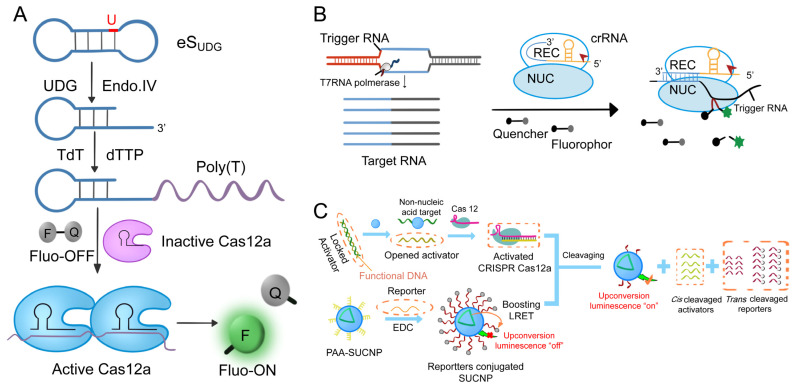
CRISPR protein analysis system with fluorescence sensor. (**A**) Schematic of the TdT-combined CRISPR/Cas12a method for UDG activity assay [108]. (**B**) Schematic of a Cas13a/crRNA-mediated, RNA-triggered signal amplification system following DNA transcription [134]. (**C**) Schematic of a DNA-regulated CRISPR/Cas12a biosensor designed to detect non-nucleic acid targets. This biosensor combines enhanced upconversion LRET with luminescence amplification facilitated by a biomimetic PC chip [139].

**Table 2 ijms-24-13447-t002:** Summary of biosensing assays based on the CRISPR systems.

Classification	Method	Cas Protein Based	Amplification	Analysis	Readout	Sensitivity	Quantification	Time	Portability	Ref.
nucleic acid analysis	NASBACC	Cas9	NASBA	DNA	Fluorescence	aM	N	~3 h	N	[92]
	CAS-EXPAR	Cas9	EXPAR	IncRNA	Fluorescence	aM	N	~1 h	N	[102]
	RACE	Cas9	RCA	RNA	colorimetric	fM	Y	~4 h	N	[103]
	E-CRISPR	Cas9/Cas12a	N	DNA	electrochemistry	pM	Y	~1 h	N	[104]
	HOLMES	Cas12a	PCR; RT-PCR	DNA/RNA	Fluorescence	aM	N	~1 h	N	[90]
	HOLMESv2	Cas12b	LAMP	DNA/RNA	Fluorescence	aM	Y	~1 h	Y	[105]
	DETECTR	Cas12a	RPA	DNA	Fluorescence	aM	N	~2 h	N	[24]
	Ultrafast visual detection platform	Cas12a	PCR	DNA	Fluorescence	1.28 copies	N	~10 min	Y	[97]
	CRISPR/hybrid Cas	Cas12a, Cas13a	N	DNA/RNA	Fluorescence	10 viral copies/μL	Y	<2 h	N	[106]
	SHERLOCK	LwCas13a	RPA	DNA/RNA	Fluorescence	aM	N	2–5 h	Y	[30]
	HUDSON + SHERLOCK	LwCas13a	RPA	RNA/DNA	Fluorescence	aM	N	~2 h	Y	[91]
	DETECTR-Cas12f	Cas12f	RPA	DNA	Fluorescence	aM	N	~2 h	N	[107]
protein analysis	TdT-combined CRISPR/Cas12a amplification	Cas12	N	UDG	Fluorescence	uM	Y	~4 h	N	[108]
	CLISA	Cas13	N	IL-6, VEGF	Fluorescence	ng/Ml	Y	-	N	[109]
	E-CRISPR	Cas12a	LAMP	Protein	electrochemistry	nM	Y	~1 h	Y	[110]
ATP analysis	CRISPR-LbCas12a biosensor	LbCas12a	N	ATP	Fluorescence	μM	Y	40 min	Y	[100]
	ASD-Cas12a	LbCas12a	SDA	ATP	Fluorescence	μM	Y	~20 min	N	[101]
	Molecular Radar	Cas12a	N	ATP	Fluorescence	nM	Y	~25 min	Y	[99]
	BPE-ECL	Cas12a	N	ATP	electrochemistry	nM	Y	~2 h	N	[111]
	BA-CASLFA	Cas12a	N	ATP	Fluorescence	μM	Y	~26 min	N	[112]
	dsDNA-ZIF-90@Ag_3_AuS_2_@Fe_3_O_4_ nanoplatform	Cas12a	N	ATP, ADP	Fluorescence	nM	Y	~30 min	N	[113]

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
