# Peer review of "CRISPR/Cas-Based Techniques for Live-Cell Imaging and Bioanalysis"

_ijms, 2023, doi:10.3390/ijms241713447_

Round 1

Reviewer 1 Report

The manuscript by Huang et al. attempts to conduct a literature review focused on CRISPR/Cas-based bioimaging and bioanalysis techniques. While the topic is intriguing to a broader audience, the writing and literature review presented in the current draft are challenging to comprehend and require extensive editing throughout the entire text before being considered for further review process and publication.

Here are some suggestions:

1.      The title needs revision –

CRISPR-Cas system for bioimaging and bioanalysis to –

CRISPR/Cas-based techniques for live-cell imaging and bioanalysis

2.      The abstract appears incomplete, and several phrases have not been used correctly. The authors should carefully review and revise the abstract to ensure that it provides a clear and concise summary of the research. Additionally, authors need to make efforts to accurately use the phrases to enhance the overall quality of the abstract and make it more understandable to readers.

e.g., ease of use or easy to use?

In this summary, we summarize or In this review, we summarize?

3.      The figure quality and representation in the manuscript do not meet scientific standards.

4.      Table 1 appears to be poorly organized, and there are concerns regarding the correctness of several terms mentioned in the table. For example, dCas9-eGPF, sgRNA-FP, and so on. Full form of abbreviated terms needs to be provided in the footnote.

5.      It is pretty challenging to provide point-by-point comments due to consistent grammar and word usage issues in almost every sentence. I will offer specific phrases as examples to draw the attention of the authors to areas that need improvement.-

-          Line 39- boanalysis or bioanalysis? In this overview or In this review? we emphasis or we emphasize? research advance or research advances?

-          CRISPR/Cas, CRISPR or CRISPR-Cas or CRISPR-Cas9?

-          Figure 1. Fundamental components of (A) CRISPR-Cas9), (B) -Cas12a, (C) -Cas12f, and (D) -43 Cas13a systems[3].???

-          Line 48- fights off the invasion of bacteria and viruses?

6.      The authors should recheck the cited references for accuracy.  

For instance, Line 219- In 1997, fluorescent RNA was microinjected into living cells to start the process of developing RNA imaging of living cells[55]….

Reference [55] is - Requirement for Drosophila cytoplasmic tropomyosin in oskar mRNA localization | Nature’. https://www.nature.com/articles/377524a0 (accessed May 03, 2023).?

The writing and literature review presented in the current draft is challenging to comprehend and require extensive editing throughout the entire text before being considered for further review process and publication.

Author Response

Thanks for the valuable suggestions for our manuscript. We have revised and answered the questions one by one in the file that we uploaded. Hopefully all our answers could erase your concern for the potential publication of our manuscript. PS: All the revised text were written in red in the updated manuscript.

Reviewer: 1

1.Q: The title needs revision –CRISPR-Cas system for bioimaging and bioanalysis to –CRISPR/Cas-based techniques for live-cell imaging and bioanalysis

Author reply: Thanks for the suggestion. We have changed the title to“CRISPR/Cas-based techniques for live-cell imaging and bioanalysis”.

2.Q The abstract appears incomplete, and several phrases have not been used correctly. The authors should carefully review and revise the abstract to ensure that it provides a clear and concise summary of the research. Additionally, authors need to make efforts to accurately use the phrases to enhance the overall quality of the abstract and make it more understandable to readers.

e.g., ease of use or easy to use?

In this summary, we summarize or In this review, we summarize?

Author reply: Thanks for the suggestion.We have made major changes to the abstract:

CRISPR/Cas systems have been found widespread applications in gene editing due to their high accuracy, high programmability, ease to use, and affordability. Benefiting from the cleavage properties (trans- or cis-) of cas enzymes, the scope of CRISPR/Cas systems has expanded beyond gene editing and has been utilized in various fields, particularly in live-cell imaging and bioanalysis. In this review, we summarize some fundamental working mechanisms and concepts of the CRISPR/Cas systems, describe the recent advances and design principles of CRISPR/Cas mediated techniques employed in live-cell imaging and bioanalysis, highlight the main applications in the imaging and biosensing of a wide range of molecular targets, as well as discuss the challenges and prospects of CRISPR/Cas systems in live-cell imaging and biosensing. By illustrating the imaging and bio-sensing processes, we hope this review will guide the best use of the CRISPR/Cas in imaging and quantifying biologically and clinically elements and inspire new ideas for better tool design in live-cell imaging and bioanalysis.

3.Q The figure quality and representation in the manuscript do not meet scientific standards.

Author reply: Thanks for the suggestion.We have redrawn the images and improved their clarity.

4.Q Table 1 appears to be poorly organized, and there are concerns regarding the correctness of several terms mentioned in the table. For example, dCas9-eGPF, sgRNA-FP, and so on. Full form of abbreviated terms needs to be provided in the footnote.

Author reply: Thanks for the suggestion. We have added the abbreviation to the abbreviation list at the end of the article. We have reorganized the language of Table 1.

Changes made:

Replaced " dCas9-eGPF " with " dCas9-EGPF” 

Replaced "SaCas9, SpCas9" with "dSaCas9-EGFP,dSpCas9-mCherry” 

Replaced "sgRNA-FPs" with "sgRNA-aptamer-FP”  

5.Q It is pretty challenging to provide point-by-point comments due to consistent grammar and word usage issues in almost every sentence. I will offer specific phrases as examples to draw the attention of the authors to areas that need improvement.

- Line 39- bioanalysis or bioanalysis? In this overview or In this review? we emphasize or we emphasize? Research advances or research advances?

- CRISPR/Cas, CRISPR or CRISPR-Cas or CRISPR-Cas9?

- Figure 1. Fundamental components of (A) CRISPR-Cas9, (B) -Cas12a, (C) -Cas12f, and (D) -Cas13a systems[3].

- Line 48- fights off the invasion of bacteria and viruses?

Author reply:

Sincerely thank you for the advice, in conjunction with the second reviewer's revisions, we have made comprehensive changes to the overall grammar and wording of the article. Here is a list of the changes made to specific examples of phrases.

  1. We have uniformly revised “CRISPR-Cas” to “CRISPR/Cas” throughout the entire manuscript. The instances where “CRISPR” was mistakenly used instead of “CRISPR/Cas” have been corrected. (Line737,740)
  2. Replaced“Figure 1. Fundamental components of (A) CRISPR-Cas9, (B) -Cas12a, (C) -Cas12f, and (D) -Cas13a systems[3].”with“Fundamental Components of four CRISPR/Cas Systems. (A) CRISPR/Cas9.(B) CRISPR/ Cpf1.(C) CRISPR/Cas14. (D)CRISPR/Cas13a system.” (Line 49-50)
  3. Replaced " fights off the invasion of bacteria and viruses” with“fights off the invasion of viruses.”(Line 53)

6.Q The authors should recheck the cited references for accuracy. For instance, Line 219- In 1997, fluorescent RNA was microinjected into living cells to start the process of developing RNA imaging of living cells [55]...

Reference [55] is - Requirement for Drosophila cytoplasmic tropomyosin in oskar mRNA localization | Nature”.

Author reply: Thanks for the suggestion. We have reviewed and revised the references throughout the manuscript.

  1. –All the changed text has been writen in red in the manuscript of the updated version.

Reviewer 2 Report

In the review titled “CRISPR-Cas system for bioimaging and bioanalysis“ Shuo Huang, Rui Dai, et al. in detail describe the use of CRISPR-Cas systems for imaging and as reporters. The manuscript contains introduction, CRISPR-Cas application in imaging, in bioanalysis and the outlook.

The manuscript is quite substantial in terms of covering application areas and covering studies of diverse groups. Authors discuss most recent discoveries in CRISPR-Cas applications such as use of cas7-11 or craspase. The manuscript covers the vast variety of CRISPR-Cas applications for imaging and sensors/reporters in a comprehensive way.

I would ask for better polish for the language of the manuscript, because some statements in the text make no sense (see below).

I don’t see any major problem with the text except for:

1. Introduction - I would suggest rewriting it or seriously work on English in this section. This section covers the mechanism of actions of CRISPR-Cas effectors, however it doesn’t cover imaging or bioanalysis. I suggest shortening the CRISPR-Cas part and adding a small section on how it solves the problems in the imaging field.

2. There is an inconsistency in terminology regarding the CRISPR-Cas systems. Authors use mostly Cas12a name for Cpf1 gene, however they use Cas14 for Cas12f. I would suggest using the initial gene terminology (Cpf1 for cas12a, and cas14 as is) or use the names from the accepted CRISPR-Cas classification - Cas12a and Cas12f (for Cas14) (Makarova, K.S., Wolf, Y.I., Iranzo, J. et al. Evolutionary classification of CRISPR–Cas systems: a burst of class 2 and derived variants) just to be consistent in naming. 

Here are some specific issues in the text:

Line 23: CRISPR is mostly related to the array and it is not repeated. “created” is not the correct word in link to CRISPR sequence. CRISPR is the array, CRISPR-Cas is the defense system, so “defense mechanism” addressed to CRISPR is incorrect. CRISPR-Cas does not serve “against the proliferation of bacteria” (only in anecdotal cases).

Line 30-31: “gene editing” repeated twice with the same meaning.

Line 32: Meaning of this sentence is not clear. CRISPR-Cas is harnessed for genome editing, not for investigation of genome editing. 

Line 34: references needed.

Line 48: “two parts” there are three parts – Cas1-Cas2/Cas4 adaptation module is missing. Restructure the sentence if you don’t need this information in the text

Line 50-51: “bacteria can intercept 50 pieces of viral DNA and integrate them into their genomes” – authors talking about spacers, but this sentence sounds like they are talking about prophages. 

Line 52: “viral DNA fragments” – not clear there these fragments are coming from in the text, authors talking about spacers. CRISPR arrays are transcribed in precrRNA, then this RNA cut into crRNA containing spacer and repeats. Repeat part of crRNA binds to the cas9, not the “viral DNA fragment”

Line 64: “CRISPR-Cas9” other genes mentioned as well in this review, so “CRISPR-Cas”

Line 65: “CRISPR-Cas12” – should be CRISPR-Cas12a, there are many cas12 genes

Line 67: “crucial” not supported in the text

Line 72: Cas9 uses tracrRNA, cas12a don’t

Line 76: expands our understanding – not explained in the text, should be dropped

Line 78:  in 1.3 authors mixing Type V and Type VI systems, I suggest moving cas12f (cas14) to 1.2 to cas12a, or make a separate entry

Line 81-82: “precise nucleic acid detection” no logical connection to the previous sentence where RNA detection is described, I suggest to rearrange sentences.

Line 90: “require an accessory protein” not clear what authors mean here

Line 95-97: “we anticipate” why? 

Line 141-143: reference needed

Line 161: “significantly” – the use of this word is not clear

Line 168: “tetrameric aza ring” not clear what it is

Line 196: “often fail” reference needed

Line 233: “PAMmer” short description would help here 

Figure 3: A,B,C,D subfigure titles missing on the figure

Line 295-296: reference needed

Line 413-420: seems to miss results from 2.2.2 and 2.2.3 

Line 494: “Cas12” – should be Cas12a/b, Cas12 is large gene family

Line 566- 570: references needed

Line 570: “Like Cas12, Cas14a” - Cas14 is Cas12 (Makarova, K.S., Wolf, Y.I., Iranzo, J. et al. Evolutionary classification of CRISPR–Cas systems: a burst of class 2 and derived variants.)

Line 648: “high targeting” – not clear

Line 732: “CRISPR” – should be CRISPR-Cas

See my comments in the previous section.

Author Response

Thanks for the valuable suggestion. We have revised the questions one by one in the uploaded the file. And hopefully the answers could erase the concern for the potential publication of our manuscript. 

PS: All the revised text were written in red in the updated manuscript.

Reviewer: 2

Recommendation: Publish after minor revision.
1. Q: Introduction - I would suggest rewriting it or seriously work on English in this section. This section covers the mechanism of actions of CRISPR-Cas effectors, however, it doesn”t cover imaging or bioanalysis. I suggest shortening the CRISPR-Cas part and adding a small section on how it solves the problems in the imaging field.

Author reply: Thanks for the suggestion. In the introduction section, we have included how the CRISPR-Cas system is utilized for cellular imaging. (Line67-71, Line84-89, Line 103-106)

  1. Q: There is an inconsistency in terminology regarding the CRISPR-Cas systems. Authors use mostly Cas12a name for Cpf1 gene, however, they use Cas14 for Cas12f. I would suggest using the initial gene terminology (Cpf1 for cas12a, and cas14 as is) or using the names from the accepted CRISPR-Cas classification - Cas12a and Cas12f (for Cas14) (Makarova, K.S., Wolf, Y.I., Iranzo, J. et al. Evolutionary classification of CRISPR–Cas systems: a burst of class 2 and derived variants) just to be consistent in naming.

Author reply: Thanks for the suggestion.  We used “Cas12a” and “Cas12f” for the whole manuscript,

  1. Q: Line 23: CRISPR is mostly related to the array and it is not repeated. “created” is not the correct word in link to CRISPR sequence. CRISPR is the array, CRISPR-Cas is the defense system, so “defense mechanism” addressed to CRISPR is incorrect. CRISPR-Cas does not serve “against the proliferation of bacteria” (only in anecdotal cases).

Author reply: Thanks for the suggestion. We have changed “CRISPR is a repeated sequence created by the prokaryotic genome and serves as a defense mechanism against the proliferation of bacteria and viruses throughout the evolutionary history of life” to “CRISPR is a sequence discovered within prokaryotes, and when combined with Cas proteins, it forms a defense system that counters the proliferation of viruses throughout the evolutionary history of life.”(Line29-31)

  1. Q: Line 30-31: “gene editing” repeated twice with the same meaning

Author reply: Thanks for the suggestion. The excess “gene editing” has been removed.(Line36)

  1. Q: Line 32: Meaning of this sentence is not clear. CRISPR-Cas is harnessed for genome editing, not for investigation of genome editing.

Author reply: Thanks for the suggestion. We have changed 'investigate genome editing' to 'develop genome editing technologies'.(Line36)

  1. Q: Line 34: references needed.

Author reply: Thanks for the suggestion. Now supplementary references with additional explanations have been added.(Line36-39)

  1. Q: Line 48: “two parts” there are three parts – Cas1-Cas2/Cas4 adaptation module is missing. Restructure the sentence if you don”t need this information in the text.

Author reply: Thanks for the suggestion. We change this sentence to “The system is composed of three key elements: CRISPR, which consists of repetitive DNA sequences; Cas9, an endonuclease responsible for DNA cleavage; and the Cas1-Cas2 or Cas4 proteins, vital components for gathering and storing viral DNA.”  (Line53-62)

  1. Q: Line 50-51: “Bacteria can intercept 50 pieces of viral DNA and

integrate them into their genomes” – authors talking about spacers, but this sentence sounds like they are talking about prophages.

Author reply: Thanks for the suggestion. We reorganized this sentence to make the article clear.(Line53-59)

9.Q: Line 52: “viral DNA fragments” – not clear there these fragments are coming from in the text, authors talking about spacers.CRISPR arrays are transcribed in precrRNA, then this RNA cut into crRNA containing spacer and repeats. Repeat part of crRNA binds to the cas9, not the “viral DNA fragment”.

Author reply: Thanks for the suggestion. We changed the sentence“When a virus invades a bacterial cell, the bacteria can intercept pieces of viral DNA and integrate them into their genomes[14]. Bacterial cells transcribe these viral DNA fragments into a piece of CRISPR RNA (crRNA), which binds to the Cas9 enzyme into complexes. These complexes can be coupled with viral DNA sequences that are identi-cal, enabling the Cas9 enzyme to cleave the DNA and thwart viral replication and in-fection[15]” with “When a bacterial cell is invaded by virus, it integrates a portion of the viral DNA into its own CRISPR region, forming new spacer sequences. This process is mediated by Cas1-Cas2 or Cas4 proteins.Through this mechanism, bacteria are able to acquire and store genetic information about the virus, which is then incorporated into CRISPR RNA (crRNA)[15]. The crRNA forms a complex with the Cas9 enzyme. These com-plexes can bind to the corresponding viral DNA sequences, allowing Cas9 to cleave the DNA and prevent viral replication and infection[16].”(Line53-62)

  1. Q: Line 64: “CRISPR-Cas9” other genes mentioned as well in this

review, so “CRISPR-Cas”.

Author reply: Thanks for the suggestion. “CRISPR-Cas9” has been replaced by “CRISPR/Cas”.(Line74)

  1. Q: Line 65: “CRISPR-Cas12” – should be CRISPR-Cas12a, there

are many cas12 genes.

Q: Line 78: in 1.3 authors mixing Type V and Type VI systems, I

suggest moving cas12f (cas14) to 1.2 to cas12a, or make a separate entry.

Author reply: Thanks for the suggestion. We have moved the introduction to Cas14, originally located in section 1.4, to section 1.2, where we now introduce both the CRISPR-Cas12a and CRISPR-Cas14 systems.(Line 106,90-95)

  1. Q: Line 67: “crucial” not supported in the text.

Author reply: Thanks for the suggestion.  “Crucial” has been replaced by “significant”. (Line77)

  1. Q: Line 72: Cas9 uses tracrRNA, cas12a doesn”t.

Author reply: Thanks for the suggestion. We have deleted the sentence “Like Cas9” to avoid any inaccuracies. (Line 79)

  1. Q: Line 76: expands our understanding – not explained in the text,

should be dropped.

Author reply: Thanks for the suggestion. We have deleted this sentence “This unique interference mechanism expands our understanding of the CRISPR-Cas system and enhances its applications in genome editing.”(Line 94)

  1. Q: Line 81-82: “precise nucleic acid detection” has no logical connection to the previous sentence where RNA detection is described, I suggest rearranging sentences.

Author reply: Thanks for the suggestion. We have removed “This high-fidelity identification technique holds great potential for precise nucleic acid detection” and replaced “Several CRISPR assays have been developed to distinguish viral strains and perform fine-typing analysis, showcasing the versatility of this approach” with “Detection methods based on CRISPR-Cas13 have been developed to differentiate virus strains and perform fine-typing analysis, showcasing the versatility of this system” to maintain the coherence of the article. (Line90-101)

  1. Q: Line 90: “require an accessory protein” not clear what the authors mean here.

Author reply: Thanks for the suggestion.Require an accessory protein” has been replaced by “need the help of accessory proteins for guidance” (Line92)

  1. Q: Line 141-143: reference needed.

Author reply: Thanks for the suggestion. We added the reference “[53]M. E. Tanenbaum, L. A. Gilbert, L. S. Qi, J. S. Weissman, and R. D. Vale, “A protein-tagging system for signal amplification in gene expression and fluorescence imaging”, Cell, vol. 159, no. 3, pp. 635–646, Oct. 2014, doi: 10.1016/j.cell.2014.09.039.” was added. (Line 152)

  1. Q: Line 161: “significantly” – the use of this word is not clear.

Author reply: Thanks for the suggestion. We use “commonly” for “significantly”.(Line167)

  1. Q: Line 168: “tetrameric aza ring” not clear what it is.

Author reply: Thanks for the suggestion. “Tetrameric aza ring” has been replaced by the “four-membered azetidine rings”. (Line 174)

  1. Q: Line 196: “often fail” reference needed.

Author reply: Thanks for the suggestion. We have reorganized this paragraph “However, effectively labeling the internal viral components without altering the virus envelope and capsid remains a challenge, and existing strategies are not applicable to most viruses.”,and added the corresponding references “[43] Y.-B. Yang et al., “Single Virus Tracking with Quantum Dots Packaged into Enveloped Viruses Using CRISPR”, Nano Lett., vol. 20, no. 2, pp. 1417–1427, Feb. 2020, doi:10.1021/acs.nanolett.9b05103.(Line199-201)”.

  1. Q: Line 233: “PAMmer” short description would help here.

Author reply: Thanks for the suggestion. We have added the full name and meaning of “PAMer”in the abbreviation glossary.

PAMmer: PAM-presenting DNA oligonucleotide. The deoxyribonucleotide sequence works as a PAM site for the Cas9 protein to cleave using its endonuclease activity. (Line 829)

  1. Q: Figure 3: A, B, C, D subfigure titles missing on the figure.

Author reply: Thanks for the suggestion. The order of A, B, and C has been marked on the picture. (Figure 3)

  1. Q: Line 295-296: reference needed.

Author reply: Thanks for the suggestion.

References have been added as below:

[68]      J. M. de la Fuente and C. C. Berry, ‘Tat peptide as an efficient molecule to translocate gold nanoparticles into the cell nucleus’, Bioconjug. Chem., vol. 16, no. 5, pp. 1176–1180, 2005, doi: 10.1021/bc050033.(Line )

[69]      S. L. Lo and S. Wang, ‘An endosomolytic Tat peptide produced by incorporation of histidine and cysteine resi-dues as a nonviral vector for DNA transfection’, Biomaterials, vol. 29, no. 15, pp. 2408–2414, May 2008, doi: 10.1016/j.biomaterials.2008.01.031.

  1. Q: Line 413-420: seems to miss results from 2.2.2 and 2.2.3.

Author reply:  Thanks for the suggestion. Here, we reintroduce some application cases of other Cas proteins for RNA imaging, aiming to enhance targeting precision.(Line415-428)

  1. Q: Line 494: “Cas12” – should be Cas12a/b, Cas12 is large gene family.

Author reply: Thanks for the suggestion.  “Cas12” has been replaced by “Cas12a/b” (Line499)

  1. Q: Line 566- 570: references needed.

Author reply: Thanks for the suggestion. [130]       B. Zhou et al., “CRISPR/Cas14 provides a promising platform in facile and versatile aptasensing with improved sensitivity”, Talanta, vol. 254, p. 124120, Mar. 2023, doi 10.1016/j.talanta.2022.124120. (Line579-581)

  1. Q: Line 570: “Like Cas12, Cas14a” - Cas14 is Cas12 (Makarova, K.S., Wolf, Y.I., Iranzo, J. et al. Evolutionary classification of CRISPR–Cas systems: a burst of class 2 and derived variants.).

Author reply: Thanks for the suggestion. We deleted the sentence “Like Cas12”, and added a reference after this sentence.(Line 572-573)

[128]    K. S. Makarova et al., “Evolutionary classification of CRISPR-Cas systems: a burst of class 2 and derived variants”, Nat. Rev. Microbiol., vol. 18, no. 2, pp. 67–83, Feb. 2020, doi: 10.1038/s41579-019-0299-x.

  1. Q: Line 648: “high targeting” – not clear.

Author reply: Thanks for the suggestion.  “high targeting” has been replaced by “high target specificity”(Line 648)

  1. Q: Line 732: “CRISPR” – should be CRISPR/Cas.

Author reply: Thanks for the suggestion. “CRISPR” has been replaced by “CRISPR/Cas”. (Line736,739,740)

  1. –All the changed text has been writen in red in the manuscript of the updated version.

Round 2

Reviewer 1 Report

The revised version is significantly improved and can be considered for publication with minor text and grammar changes.

The revised version is significantly improved and can be considered for publication with minor text and grammar changes.